# Adaptive Subset-Based Digital Image Correlation for Fatigue Crack Evaluation

## Myung Soo Kang and Yun-Kyu An *

Department of Architectural Engineering, Sejong University, Seoul 05006, Korea; kms35954@sju.ac.kr
* Correspondence: yunkyuan@sejong.ac.kr; Tel.: +82-2-6935-2426

**Abstract:** This paper proposes a fatigue crack evaluation technique based on digital image correlation (DIC) with statistically optimized adaptive subsets. In conventional DIC analysis, a uniform subset size is typically utilized throughout the entire region of interest (ROI), which is determined by experts' subjective judgement. The basic assumption of the conventional DIC analysis is that speckle patterns are uniformly distributed within the ROI of a target image. However, the speckle patterns on the ROI are often spatially biased, augmenting spatially different DIC errors. Thus, a subset size optimization with spatially different sizes, called adaptive subset sizes, is needed to improve the DIC accuracy. In this paper, the adaptive subset size optimization algorithm is newly proposed and experimentally validated using an aluminum plate with sprayed speckle patterns which are not spatially uniform. The validation test results show that the proposed algorithm accurately estimates the horizontal displacements of 200 μm, 500 μm and 1 mm without any DIC error within the ROI. On the other hand, the conventional subset size determination algorithm, which employs a uniform subset size, produces the maximum error of 33% in the designed specimen. In addition, a real fatigue crack-opening phenomenon, which is a local deformation within the ROI, is evaluated using the proposed algorithm. The fatigue crack-opening phenomenon as well as the corresponding displacement distribution nearby the fatigue crack tip are effectively visualized under the uniaxial tensile conditions of 0.2, 1.0, 1.4 and 1.7 mm, while the conventional algorithm shows local DIC errors, especially at crack opening areas.

**Keywords:** fatigue crack evaluation; digital image correlation; adaptive subset size; statistical optimization; automated subset size determination

## 1. Introduction

A fatigue crack caused by cyclic loading is one of the most critical damage types in metallic structures because it may cause plastic deformation or abrupt structural failure even below the yield strength. However, the technical challenge in fatigue crack evaluation is that a fatigue crack cannot typically be observed by the naked eye. To effectively investigate the fatigue crack, a number of nondestructive evaluation (NDE) techniques—such as ultrasonic [1–3], infrared thermography [4–6], eddy current [7–9], shearography [10–12], radio-frequency identification [13–15], vision-based inspection [16–18] and digital image correlation (DIC) [19–21]—have been developed. Among various NDE techniques, DIC is one of the simplest and most promising optical assessment tools for fatigue crack evaluation, because DIC is able to intuitively visualize and trace the minute deformation of a target structure in a pixel unit. To accurately evaluate the fatigue crack using DIC, users should carefully consider various factors, such as non-parallel charge-coupled device [22], measurement noise [23,24], correlation criterion [25], shape function [26–29] and subset size [30–34]. Among them, the subset size is one of the most critical factors in terms of DIC accuracy, because the partial deformation of a target surface is separately tracked by speckle pattern features within each subset. The smaller subset size is

typically able to achieve the higher DIC accuracy by increasing the spatial resolution. On the other hand, an excessively small subset size often inversely increases the DIC errors, because they may not contain enough distinctive speckle pattern features within each subset for a proper DIC analysis.

Although commercial DIC software—such as StrainMaster from LaVision [35], GOM Correlate from GOM [36] and VIC-2D from Correlated Solution [37]—have been used, they commonly select the subset size depending on users' subjective judgement without a speckle pattern analysis. To determine the optimal subset size, a number of subset size determination algorithms have been proposed. Yaofeng and Pang suggested a subset entropy, calculating the sum of the absolute difference of eight neighboring points for the selection of a single subset size [30]. Pan et al. proposed a sum of the square of the subset intensity gradient (SSSIG), which evaluates the local speckle pattern intensity using a threshold value of standard deviation (SD) error within the subset for selecting a single subset size throughout the entire region of interest (ROI) [31]. Additionally, Lane et al. proposed a grey-level co-occurrence matrix (GLCM) to determine a single subset size using a critical GLCM offset concept [32]. Although the aforementioned algorithms can properly determine a single optimal subset size within the entire ROI, the following technical hurdles still remain to be overcome. First, the conventional algorithms evaluate the speckle patterns at a certain local area in the target image and determine a single subset size under the assumption of uniformly distributed speckle patterns within the entire ROI. If the speckle patterns are spatially biased, it may augment spatially different DIC errors. Furthermore, the conventional algorithms highly depend on the experts' subjective judgement or experience to determine an optimal threshold value for the subset size determination. Although dynamic subset selection algorithms to adopt various subset sizes in the ROI were recently proposed [33,34], they are not fully validated in spatially biased speckle patterns yet. Therefore, a fully automated subset size optimization study is still necessary.

To come up with the technical demand, a fully automated adaptive subset size determination algorithm is newly proposed and experimentally validated through a fatigue crack-opening evaluation with spatially biased speckle-patterned images in this study. The adaptive subset sizes are spatially different depending on the speckle pattern quality of each local subset and automatically determined by the iteration of normalized cross correlation (NCC) without experts' intervention within the entire ROI. Another superiority of the proposed algorithm over the existing algorithms is that the random measurement noises can be minimized using the combination of several images acquired with a certain time interval without deformation of a target structure. Finally, the effectiveness of the proposed adaptive subset size determination algorithm is experimentally validated using a speckle-patterned aluminum specimen with a sophisticatedly controllable scanning stage. Then, the fatigue crack-opening phenomenon, which is a local deformation on the ROI, is evaluated using a universal testing machine (UTM). In addition, the experimental results are compared with SSSIG, which is one of the most widely accepted conventional subset size determination algorithms, for further quantitative validation.

This paper is organized as follows. First, the automated adaptive subset size determination algorithm is explained. The feasibility tests of the proposed algorithm are then conducted with the spatially biased speckle-patterned aluminum specimen. Next, the fatigue crack-opening phenomenon is evaluated with adaptive subset sizes. Finally, this paper is concluded with a brief discussion.

## 2. Automated Determination Algorithm of Adaptive Subset Sizes

Figure 1 shows the overview of the automated adaptive size determination algorithm. The proposed algorithm consists of the following four steps: (1) initial parameter setting within the ROI, (2) the determination of the converging size by evaluating a matching distance, (3) the establishment of a convergence map and (4) the determination of adaptive subset sizes. The details of each step are as follows.

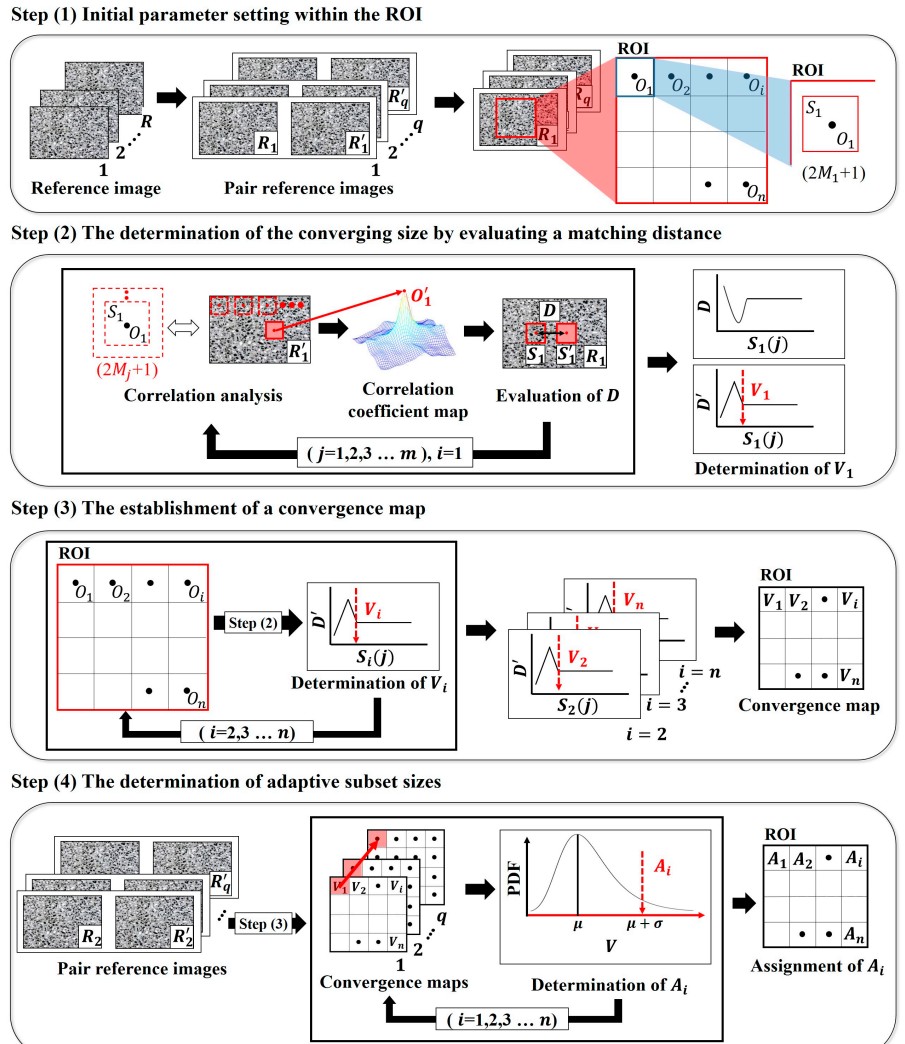

**Figure 1.** Overview of the automated size determination algorithm of adaptive subsets: $R_q$ and $R'_q$ are the pair reference images. $O_i$ is the seed point on $R_q$, and $S_i$ is the reference subset centered at $O_i$. $M_j$ is the size parameter of $S_i$, and $S_i(j)$ is $(2M_j + 1) \times (2M_j + 1)$. $S'_i$ is the matched subset of $S_i$ centered at $O'_i$. $D$ is the matching distance between $S_i$ and $S'_i$. $D'$ is the derivative of $D$. $V_i$ is the converging size and $A_i$ is the adaptive subset size. PDF is the probability density function.

Step (1) Initial parameter setting within the ROI: First, $R$ number of the reference images are taken by a digital camera at a certain time interval without any deformation of the target structure, as shown in Step (1) of Figure 1. The pair of reference images, i.e., $R_q$ and $R'_q$, are then selected via the two different combinations of the reference images. Subsequently, the ROI to be analyzed is selected within $R_q$, and the seed point $O_i$ is then spatially assigned with a certain spatial interval on the ROI. Note that the spatial interval of $O_i$ should not be larger than the minimum size of the reference subset $S_i$ to be investigated in the subsequent step, so that searching pixel missing can be avoided. Once $O_i$ is assigned in the ROI, $S_1$, centered at $O_1$, starts to be determined with the size of $(2M_1+1) \times (2M_1+1)$ to have integer pixel values.

Step (2) The determination of the converging size by evaluating a matching distance: As depicted in Step (2) of Figure 1, the NCC of $S_1$ with respect to $R'_1$ is calculated to establish the correlation coefficient ($C_{NCC}$) map. $C_{NCC}$ can be expressed by:

$$C_{NCC} = \sum_{a=-M_j}^{M_j} \sum_{b=-M_j}^{M_j} \left[ \frac{[f(x_a,y_b)-f'] \times [g(x'_a,y'_b)-g']}{\Delta f \Delta g} \right],$$

$$\Delta f = \sqrt{\sum_{a=-M_j}^{M_j} \sum_{b=-M_j}^{M_j} [f(x_a,y_b)-f']^2}, \ \Delta g = \sqrt{\sum_{a=-M_j}^{M_j} \sum_{b=-M_j}^{M_j} [g(x'_a,y'_b)-g']^2} \quad (1)$$

$$f' = \frac{1}{(2M_j+1)^2} \sum_{a=-M_j}^{M_j} \sum_{b=-M_j}^{M_j} f(x_a,y_b), \ g' = \frac{1}{(2M_j+1)^2} \sum_{a=-M_j}^{M_j} \sum_{b=-M_j}^{M_j} g(x'_a,y'_b)$$

where $f$ and $g$ represent the grayscale intensity value at spatial points $(x_a, y_b)$ and $(x'_a, y'_b)$ in $R_q$ and $R'_q$, respectively.

The pixel of the highest $C_{NCC}$ within the $C_{NCC}$ map is selected as $O'_1$ which is the center point of the matched subset $S'_1$. Physically, $S'_1$ is the most similar to $S_1$ within $R'_1$. $S'_1$ is then assigned to $R_1$, centered at $O'_1$. If there is no deformation between $R_q$ and $R'_q$, $O'_i$ and $O_i$ theoretically have the same locations on $R_q$ and $R'_q$, respectively. Next, the matching distance $D$ between $S_1$ and $S'_1$ is computed using $O_1$ and $O'_1$, which is given by:

$$D = \sqrt{(x_i - x'_i)^2 + (y_i - y'_i)^2}, \quad (2)$$

where $(x_i, y_i)$ and $(x'_i, y'_i)$ are the spatial points of $O_i$ and $O'_i$ on $R_q$ and $R'_q$, respectively.

Now, $C_{NCC}$ is iteratively calculated by increasing $S_1(j)$, i.e., $(2M_j+1) \times (2M_j+1)$. Here, $M_j$ ($j = 1,2,3 \dots m$) is the size parameter of $S_i$, and $S_i(j)$ should be smaller than the ROI. Then, $D$ can be obtained depending on $S_1(j)$. When $S_i(j)$ has small values, $D$ typically fluctuates, as shown in Step (2) of Figure 1, because the lack of distinctive speckle features within the $S_i$ makes it difficult to find the exact location of $S'_i$. On the other hand, $D$ will converge after $S_i(j)$ exceeds a certain value, which physically implies that sufficient speckle features are secured within the subset. The threshold value can be considered as the minimum converging size $V_1$, which is determined when the derivative of $D$ ($D'$) becomes 0, as shown in Step (2) of Figure 1.

Step (3) The establishment of a convergence map: As for $O_i$ ($i = 2,3 \dots n$), $V_i$ can be obtained by repeating Step (2), as described in Step (3) of Figure 1. Then, $V_i$ is assigned at the corresponding $O_i$ within the ROI, which is called the convergence map. Physically, $V_i$ in the convergence map means the minimum subset size for the proper DIC analysis with respect to each $O_i$ within the ROI.

Step (4) The determination of adaptive subset sizes: In the last step, $q$ number of the convergence maps can be obtained from $R_q$ and $R'_q$, as shown in Step (4) of Figure 1. The reason why the multiple convergence maps are used in this algorithm is that the random measurement noises caused during the image acquisition process can be minimized through averaging. If the random measurement noises are more dominant than the speckle features within a certain size of subset, $V_i$ will be increased. Thus, for each pair of reference images, $V_i$ might be different depending on the random measurement noises even at the same $O_i$. Then, $V_i$ can be assumed to follow the normal distribution because $V_i$ depends on random measurement noises. Finally, the adaptive subset size $A_i$ is statistically determined by summing the SD ($\sigma$) and mean ($\mu$) with respect to the $q$ number $V_i$ of each $O_i$.

## 3. The Feasibility Tests of Adaptive Subset Sizes

The feasibility of the proposed algorithm is experimentally validated using a speckle-patterned aluminum plate specimen, as shown in Figure 2. To quantitatively investigate the surface deformation, horizontal displacements of 200 µm, 500 µm and 1 mm are applied to the specimen using the scanning stage in this experiment. Furthermore, the test results are compared with the conventional SSSIG algorithm.

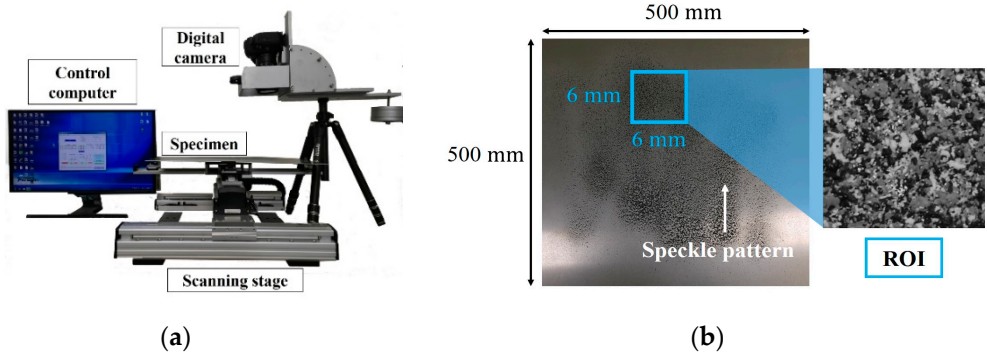

**(a)**                                                                                    **(b)**

**Figure 2.** Experimental setup: (**a**) data acquisition system, (**b**) aluminum plate specimen and the target region of interest (ROI).

Figure 2a shows the data acquisition system, consisting of the control computer, digital camera, speckle-patterned aluminum specimen and scanning stage. The overall working principle is as follows. First, several reference images are acquired from the surface of the scanning stage-mounted specimen. To avoid image distortion, the digital camera keeps the parallel aligned condition with respect to the target specimen surface. The adaptive subset sizes can be constructed within the entire ROI from the combinations of reference images. Then, the control computer sends out a signal to the scanning stage to numerically translate the specimen in the $x$ or $y$ directions, and the corresponding surface is acquired as a test image for the validation test. By repeating the specimen translation and test image acquisition over the predetermined steps, the adaptive subset sizes are validated through comparing the actual translated displacement and the displacement which calculated by the DIC. Finally, the validation test results are compared with the conventional subset size determination algorithm, i.e., SSSIG.

The digital camera employed in this system is the Canon EOS 5D Mark 4 with a 100 mm F 2.8 L macro IS USM lens. The scanning stage is able to shift the specimen along the $x$ and $y$ directions with a spatial resolution of 0.5 $\mu$m. Then, the speckle patterns are made on the aluminum specimen of $500 \times 500 \times 2$ mm$^3$ using a stone spray, as shown in Figure 2b. Here, the speckle patterns are spatially biased and intentionally designed to examine the feasibility of the proposed algorithm. The speckle pattern images of the ROI, which has $300 \times 300$ pixels ($6 \times 6$ mm$^2$) on the target specimen, are taken by the digital camera. The image resolution is $3360 \times 2240$ pixels when the working distance between the lens and the target specimen surface is 230 mm. Here, a single pixel is equivalent to 20 $\mu$m. The digital images are obtained under the normal indoor lighting condition, and the camera setting is fixed at ISO 1600, F 22 and an exposure time of 0.5 s.

First, 15 reference images without any deformation of the target specimen are acquired from the ROI with five second time intervals, and the corresponding 105 pairs of reference images are obtained in Step (1). Then, 2500 $O_i$ are assigned to the ROI with respect to the spatial interval of six pixels. For all $O_i$, the $V_i$ values are determined while increasing the $M_j$ from 3 to 27 with intervals of 1 through Step (2) and Step (3). Finally, in Step (4), 105 convergence maps are established, and $A_i$ values are determined with respect to the 105 $V_i$ values; they are then assigned at each $O_i$.

Figure 3 shows the determination results of $A_i$. $A_i$ varies from the minimum $9 \times 9$ pixels to the maximum $23 \times 23$ pixels depending on the $O_i$. Here, 91% of $A_i$ has a subset size between $11 \times 11$ pixels and $17 \times 17$ pixels, as shown in Figure 3a. It is interesting to observe that 3 of the subset are automatically determined as $25 \times 25$ pixels, meaning that the corresponding area physically does not have enough distinctive speckle features compared to the other ordinary areas. Figure 3b shows the spatially different $A_i$, which is obtained through the proposed adaptive subset size optimization algorithm within the ROI of the test specimen. Note that the subset sizes can be adaptively and automatically optimized depending on the target speckle pattern, digital camera type, image capturing condition and so on.

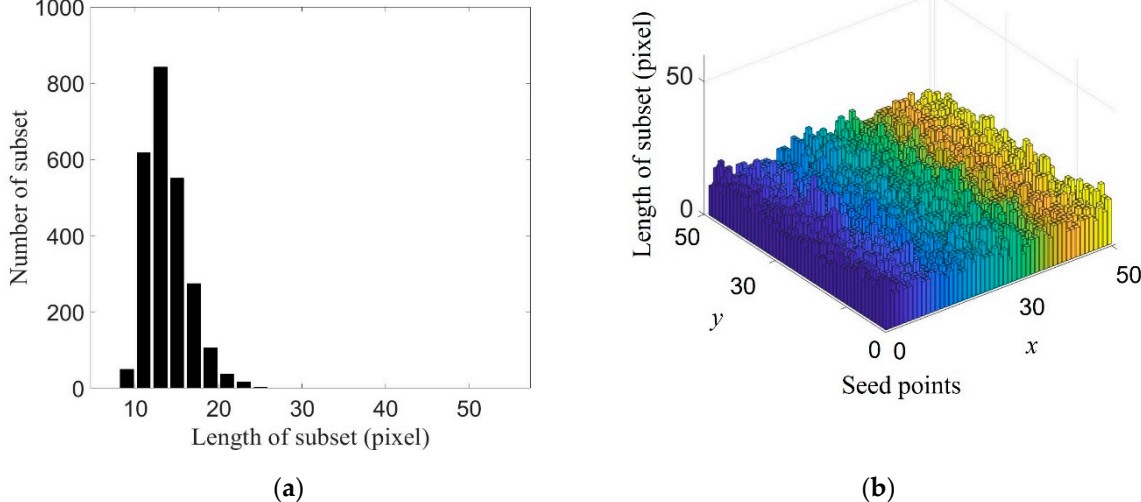

**Figure 3.** Determination results of adaptive subset sizes: (**a**) the number of adaptive subset sizes according to the subset length and (**b**) spatial distribution of the adaptive subset sizes.

To validate the optimized adaptive subset sizes, the specimen is horizontally shifted using the scanning stage and the corresponding DIC errors are computed for the entire ROI. Figure 4 shows the DIC errors when the horizontal displacements are 200 μm, 500 μm and 1 mm. The resultant images show that no error appears in the test cases, meaning that the spatially optimized subsets with distinctive features track the same speckle patterns well even when the target patterns are translated within the ROI.

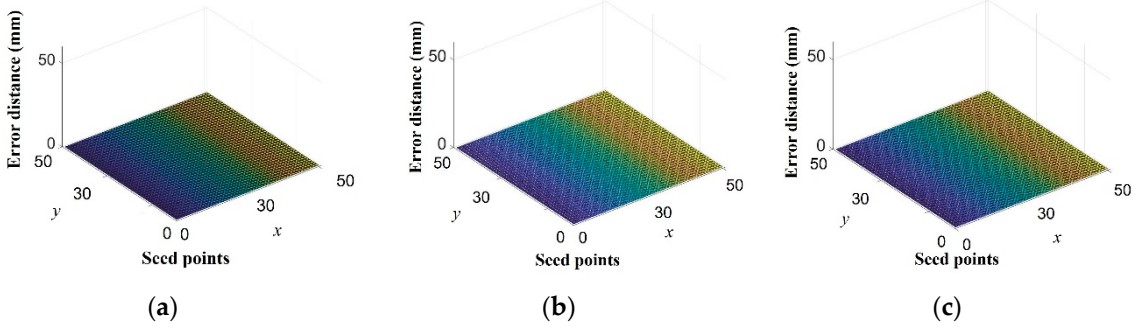

**Figure 4.** Validation test results of the adaptive subset sizes: horizontal displacements of (**a**) 200 μm, (**b**) 500 μm and (**c**) 1 mm.

In order to compare the validation results of the proposed algorithm with the conventional algorithm ones, a single subset size is determined by SSSIG. First, SSSIG selects a random location within the ROI and calculates speckle pattern gradients within the subset while increasing the subset size. Then, a single subset size that satisfies a certain threshold is determined.

According to the typical procedure of SSSIG, the two different seed points are randomly selected, as shown in Figure 5. To equivalently compare the test results, the seed points are intentionally selected among $O_i$. Then, the SD error values, i.e., $SDerror(x)$ and $SDerror(y)$, are calculated. It is given by [31]:

$$SDerror(x) \approx \left( \frac{N(\eta)}{\sum \sum (f_x)^2} \right)^{\frac{1}{2}}, \ SDerror(y) \approx \left( \frac{N(\eta)}{\sum \sum (f_y)^2} \right)^{\frac{1}{2}}, \tag{3}$$

where $N(\eta)$ is the noise variance calculated using the two images acquired with a certain time interval. $f_x$ and $f_y$ are the first-order derivatives of grayscale intensities within the subset along the $x$ and $y$ directions, respectively. The threshold of the $SDerrors$ is set to 0.005, as recommended by

Pan et al. [31], and the subset size increases from $7 \times 7$ pixels to $55 \times 55$ pixels, which is similar to the proposed algorithm.

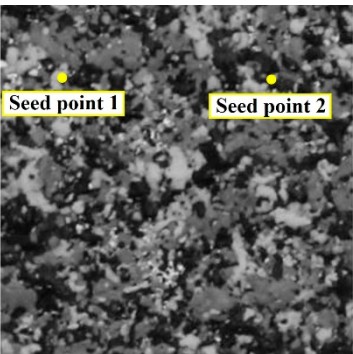

**Figure 5.** Randomly selected two different seed points within the ROI of the aluminum specimen.

Figure 6 shows the subset size determination results obtained by SSSIG. At the seed point 1, the subset size of $7 \times 7$ pixels is determined by employing the threshold value of 0.005, as shown in Figure 6a. On the other hand, at the seed point 2 the subset size of $9 \times 9$ pixels is selected, as displayed in Figure 6b. The SSSIG results obtained from the two different seed points mean that the subset size can be different depending on the spatial points and even the empirically obtained threshold value.

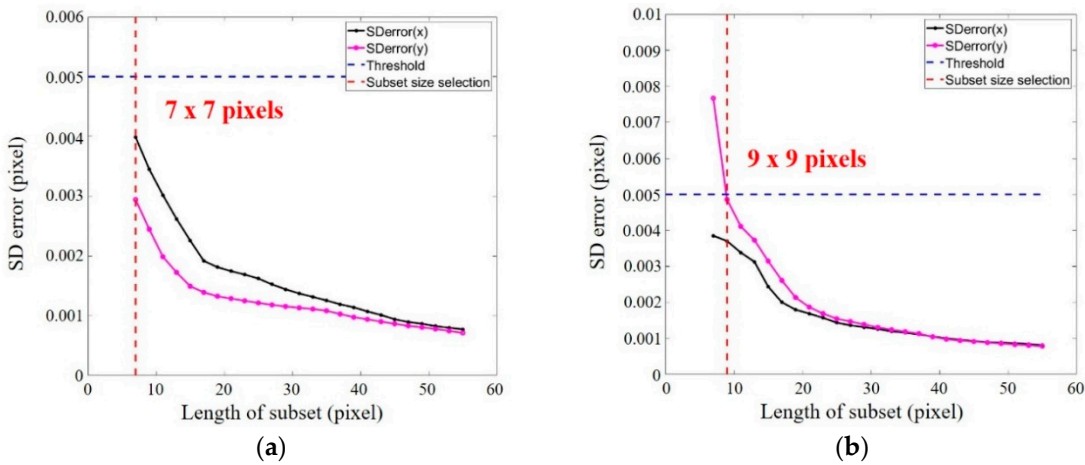

**Figure 6.** Determination of subset sizes using subset intensity gradient (SSSIG) at the randomly selected two different seed points: (**a**) seed point 1 and (**b**) seed point 2.

Similarly, the corresponding DIC errors are computed with respect to the horizontal displacement images of 200 µm, 500 µm and 1 mm. Figure 7a shows that a number of error points occur regarding the subset size of $7 \times 7$ pixels even for the 200 µm displacement. Compared with this, Figure 7a–c shows that the error points decrease as the specimen displacement increases. On the other hand, the subset size of $9 \times 9$ pixels case reveals that a certain error pattern depending on the specimen displacement cannot be observed, as shown in Figure 7d–f, while it can be clearly seen that the DIC errors randomly and fairly exist. The DIC errors are quantitatively compared in Table 1, and the maximum error of SSSIG is 33% among the test cases. As a result, it can be confirmed that spatially different random DIC errors are noticeably produced by SSSIG, while there are no DIC errors in the adaptive subset case, as shown in Figure 4.

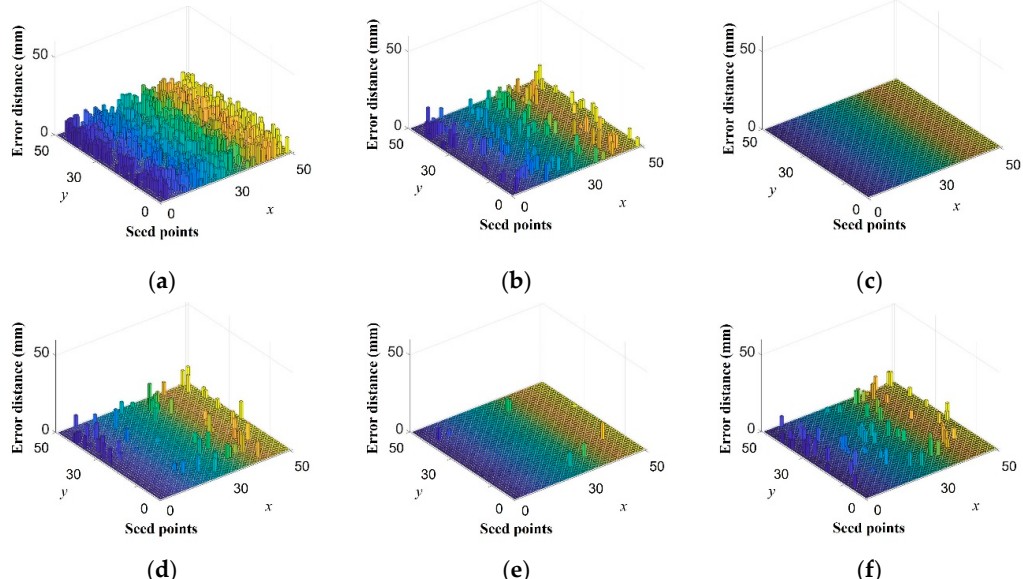

**Figure 7.** Validation test results of SSSIG: the subset size of 7 × 7 pixels with a horizontal displacement of (**a**) 200 μm, (**b**) 500 μm and (**c**) 1 mm. The subset size of 9 × 9 pixels with a horizontal displacement of (**d**) 200 μm, (**e**) 500 μm and (**f**) 1 mm.

**Table 1.** Comparison of the number of spatial error points between the adaptive subset and the SSSIG.

| Case | The Number of Spatial Error Points | | |
|---|---|---|---|
| | 200 μm | 500 μm | 1 mm |
| Adaptive subset | 0 | 0 | 0 |
| SSSIG 7 × 7 pixels | 829 | 134 | 0 |
| SSSIG 9 × 9 pixels | 67 | 8 | 71 |

## 4. Fatigue Crack-Opening Evaluation Tests

The fatigue crack-opening phenomenon, which is a typical local deformation, is experimentally traced on a speckle-patterned fatigue crack specimen, as shown in Figure 8. In this experiment, four steps of the fatigue crack-opening images are acquired under the uniaxial tensile conditions of 0.2, 1.0, 1.4 and 1.7 mm using UTM.

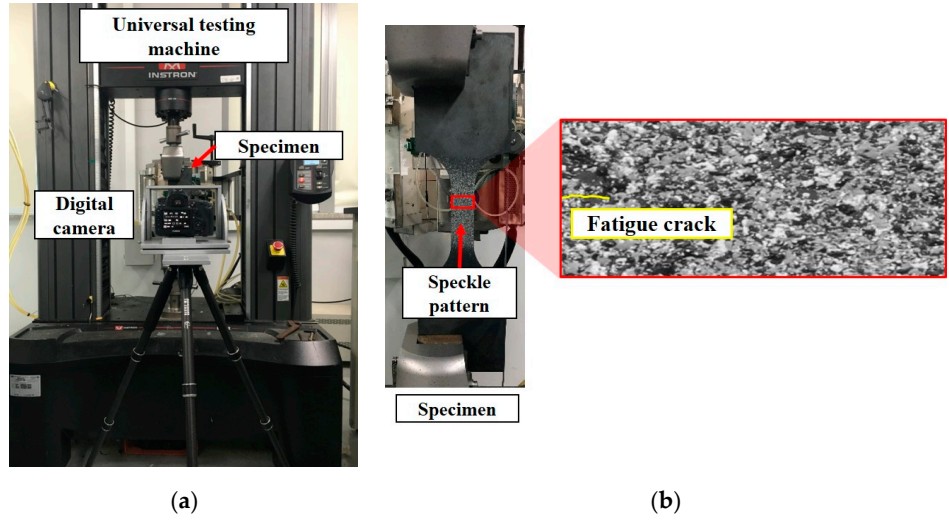

**Figure 8.** Experimental setup: (**a**) data acquisition system and (**b**) the ROI on the fatigue crack specimen.

Figure 8a shows the fatigue crack-opening test setup, consisting of the digital camera, speckle-patterned dog-bone test specimen and UTM. Figure 8b displays the zoom-in speckle-patterned image of the specimen with a real fatigue crack and the random speckle pattern similarly made by the stone spray. First, the reference images are captured from the target specimen surface, which is installed at the UTM without loading. Here, the ROI is 1000 × 400 pixels, which is equivalent to 20 × 8 mm$^2$ on the target specimen, as shown in Figure 8b. Then, the uniaxial tensile loads of 0.2, 1.0, 1.4 and 1.7 mm are applied using UTM, and the corresponding test images are sequentially acquired according to the loading steps. Note that the single pixel resolution of the captured image is 20 μm when the working distance between the lens and specimen surface is 487 mm. The speckle-patterned images are obtained under normal indoor lighting conditions and the camera settings are similarly fixed at ISO 1600, F 22, with an exposure time of 0.5 s. INSTRON 5982 UTM has a 100 kN axial force capacity with a 0.01 mm displacement resolution control.

Similar to the feasibility test, 15 reference images are acquired from the ROI without any deformation with five second time intervals; the corresponding 105 pairs of reference images are obtained according to Step (1). Then, 6250 $O_i$ are assigned to the ROI with respect to the spatial interval of seven pixels. For all $O_i$, the $V_i$ values are determined while increasing $M_j$ from 3 to 27 with intervals of 1 through Step (2) and Step (3). Finally, in Step (4), 105 convergence maps are established. Subsequently, the $A_i$ values are determined with respect to 105 $V_i$; they are then assigned at each $O_i$.

Figure 9 shows the determination results of $A_i$. $A_i$ varies from the minimum 15 × 15 pixels to the maximum 43 × 43 pixels depending on $O_i$. Here, 78.4% of $A_i$ has a subset size of between 21 × 21 and 29 × 29 pixels, as shown in Figure 9a,b, which shows the spatially different and randomly distributed subset sizes within the entire ROI of the test specimen.

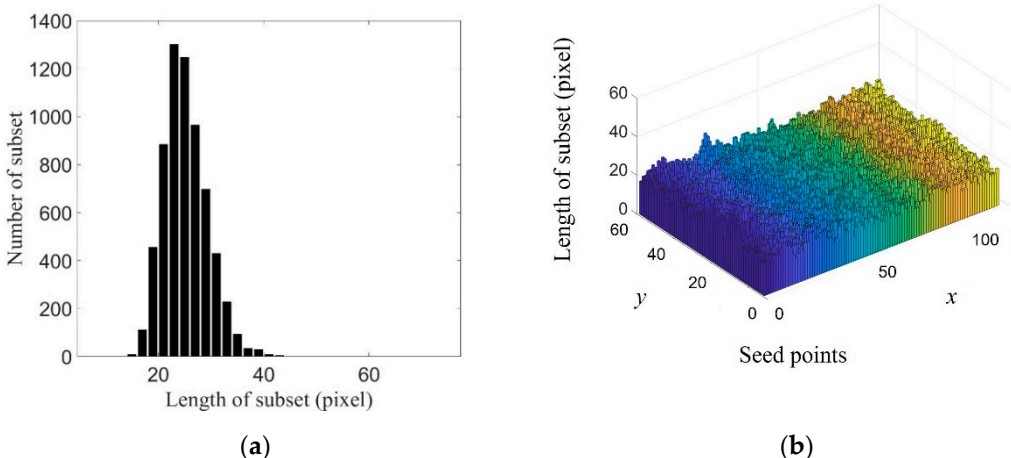

(**a**)　　　　　　　　　　　　　　　　　　　　　(**b**)

**Figure 9.** Determination results of adaptive subset sizes: (**a**) the number of adaptive subset sizes according to the subset length and (**b**) spatial distribution of the adaptive subset sizes.

Figure 10 shows the DIC analysis results with the automatically optimized adaptive subset sizes corresponding to the test images acquired under the uniaxial tensile loading conditions of 0.2 mm, 1.0 mm, 1.4 mm and 1.7 mm. The color bar of Figure 10 shows the minute displacement of the target specimen according to the loading step. Figure 10a shows that only a random displacement distribution can be observed, because a 0.2 mm displacement is too small to be observed by the determined physical subset sizes. From the 1.0 to 1.7 mm displacement steps, the corresponding displacement distributions can be clearly observed during the fatigue crack-opening, as shown in Figure 10b–d. In particular, the fatigue crack boundaries are clearly visualized in Figure 10c,d. Although the DIC error partially occurs in Figure 10d, the displacement distribution near the crack tip is successfully visualized due to the spatially different subset sizes, which are large enough to compensate the local DIC errors.

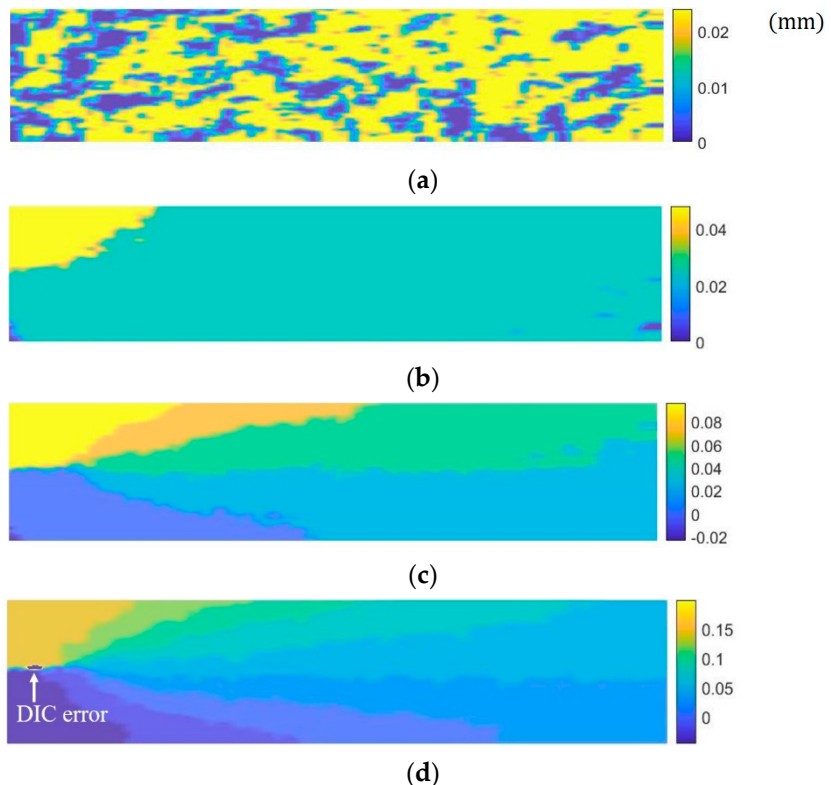

**Figure 10.** Digital image correlation (DIC) analysis results using the proposed algorithm under the uniaxial tensile loads of: (**a**) 0.2, (**b**) 1.0, (**c**) 1.4 and (**d**) 1.7 mm.

Similarly, the fatigue crack-opening phenomenon is analyzed by SSSIG. Figure 11 shows the randomly selected two seed points within the entire ROI for SSSIG.

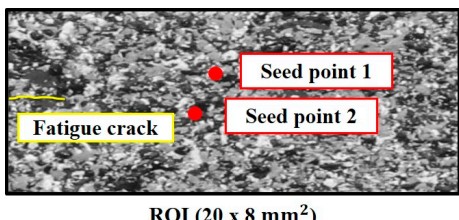

**Figure 11.** Randomly selected two seed points in the ROI of the fatigue crack specimen.

From the randomly selected two different seed points, the subset sizes of $13 \times 13$ pixels and $19 \times 19$ pixels are respectively computed throughout the entire ROI when the threshold value of 0.005 is used, as shown in Figure 12.

The corresponding DIC analysis results of subset sizes $13 \times 13$ and $19 \times 19$ pixels are shown in Figures 13a–d and 13e–h, respectively. Similarly, the resulting images of Figures 13a–d and 13e–h are clearly different from each other depending on the seed point. In particular, Figure 13a–d shows that the subset size $13 \times 13$ pixels is too small to analyze a local displacement of even 0.2 mm. On the other hand, the subset size $19 \times 19$ pixels case reveals that fatigue crack-opening is well traced until 1.0 mm, while displacement cases of over 1.4 mm produce the dominant local DIC errors, as shown in Figure 13e–h. It can be concluded that the SSSIG results highly depend on experts' subjective intervention as well as the spatial bias of the speckle pattern, making it difficult to properly analyze the fatigue crack-opening phenomenon.

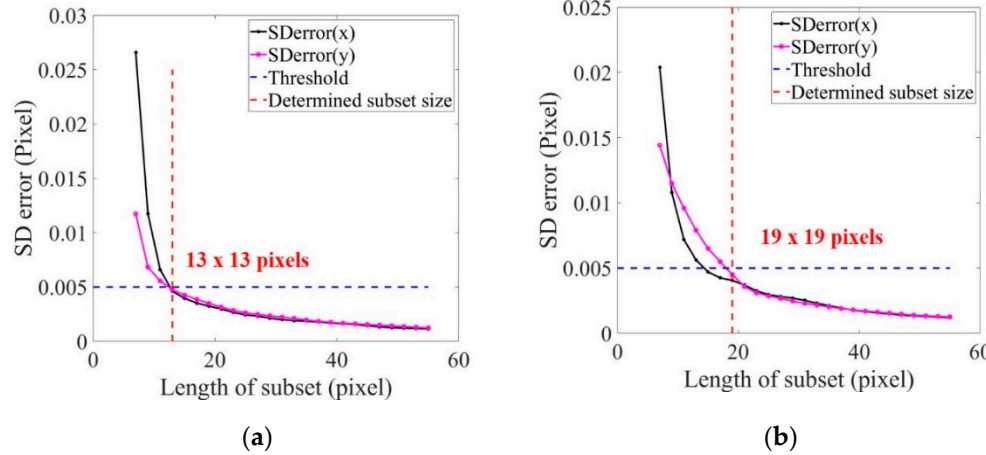

**Figure 12.** Determination of subset sizes using SSSIG at the randomly selected two different seed points: (**a**) seed point 1 and (**b**) seed point 2.

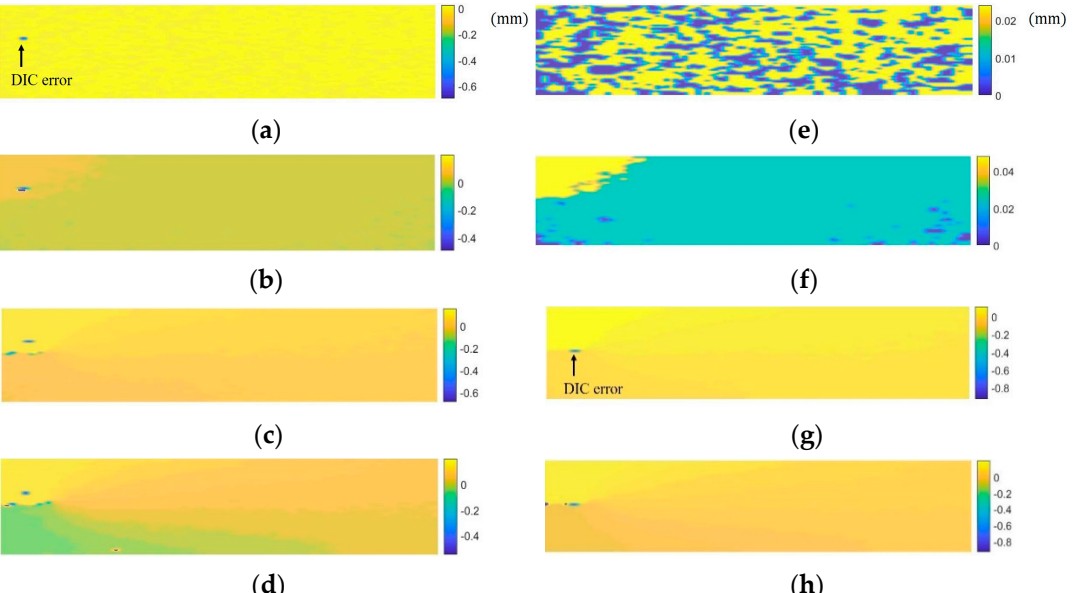

**Figure 13.** DIC analysis results using SSSIG. The subset size of 13 × 13 pixels with uniaxial tensile conditions of: (**a**) 0.2, (**b**) 1.0, (**c**) 1.4 and (**d**) 1.7 mm; the subset size of 19 × 19 pixels with uniaxial tensile conditions of (**e**) 0.2, (**f**) 1.0, (**g**) 1.4 and (**h**) 1.7 mm.

## 5. Conclusions

This paper proposed a new adaptive subset size optimization algorithm, and it was experimentally validated by tracing the global and local deformations of target specimens. In particular, the superior performance of the proposed algorithm was experimentally validated through a comparative study with one of the most widely accepted subset size determination algorithms. The validation test results revealed that the proposed algorithm automatically optimizes spatially different adaptive subset sizes in the entire region of interest without experts' subjective intervention, making it possible to successfully analyze local as well as global deformations by minimizing the local digital image correlation (DIC) errors. The proposed algorithm is able to become a promising tool for the evaluation of various local and global deformations of a target structure. In the follow-up study, the proposed algorithm will be applied to real structures such as bridges, machinery, buildings and so on. Moreover, a surface treatment-free DIC technique, which can be used as an alternative to surface-treated speckle patterns, is now being developed based on deep learning-based image feature enhancement.



**Author Contributions:** M.S.K. and Y.-K.A. conceived and designed the experiments; M.S.K. performed the validation and visualization of results; M.S.K. and Y.-K.A. wrote the paper; Y.-K.A. supervised the research. All authors have read and agreed to the published version of the manuscript.

**Funding:** This research was funded by National Research Foundation of Korea, and the Ministry of the Interior and Safety as Human Resource Development Project in Disaster Management.

**Acknowledgments:** This work was supported by the National Research Foundation of Korea (NRF) grant funded by the Korean government (MSIT) (2018R1A1A1A05078493) and the Ministry of the Interior and Safety as Human Resource Development Project in Disaster Management.

**Conflicts of Interest:** The authors declare no conflict of interest.

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
