# Peer review of "Adaptive Subset-Based Digital Image Correlation for Fatigue Crack Evaluation"

_applsci, doi:10.3390/app10103574_

Round 1
Reviewer 1 Report
The authors presented a fatigue crack evaluation technique based on digital image correlation (DIC) with statistically optimized adaptive subsets. This is an interesting improved method for proper detecting a fatigue crack. However, some points need to be clarified:
- Please update the references for the advanced technology.
- In section 2 and Figure 1, please describe the processing parameter in a consistent way.
- Equation 2 is the same with Equation 1?
- Figure 3 (b), please mention the meaning of color and value. What is difference between Figure 3(b) and 9(b)?
- In section 4, for different specimen, the setting parameters are the same with those in section 3?
- Figure 10 and 13, please mention the meaning of color bar.
Reviewer 2 Report
The manuscript presents a digital image correlation-based fatigue crack evaluation technique. Authors have proposed an adaptive subset size determination algorithm which can automatically determine iteration of NCC over entire ROI. The algorithm has been properly explained and the results were compared with the conventional SSSIG technique which showed the advantage of the proposed technique. The manuscript is well-written and structured. However, I would suggest the authors address the following issues in the manuscript:
- The manuscript has shown the advantage of the proposed technique over SSSIG technique. However, authors should add statements on what the advantage of the proposed technique is over the commercial DIC based deformation measurement system.
- Line 148 – 149: “First, several reference images are acquired from the surface of the scanning stage-mounted specimen under parallel aligned condition.”. Authors should clarify this statement. What kind of motor was used: stepper or servo? Was there a raster scan involved? How the motion of the scanner was synchronized with the images taken by the digital camera?
- Authors should add more recent literature on the different fatigue crack and vision-based damage detection techniques to make the literature studies comprehensive. I would like to suggest the following literature to the author:
- Giri, P., Kharkovsky, S. and Samali, B., 2017. Inspection of metal and concrete specimens using imaging system with laser displacement sensor. Electronics, 6(2), p.36.
- Yao, Y. and Glisic, B., 2015. Detection of steel fatigue cracks with strain sensing sheets based on large area electronics. Sensors, 15(4), pp.8088-8108.
- Giri, P., Kharkovsky, S., Samali, B. and Salama, R., 2017, November. Real-time monitoring of fatigue cracks in machine parts using microwave and laser imaging techniques. In IFToMM International Symposium on Robotics and Mechatronics (pp. 199-207). Springer, Cham.
- Camerini, C., Rebello, J.M.A., Braga, L., Santos, R., Chady, T., Psuj, G. and Pereira, G., 2018. In-Line Inspection Tool with Eddy Current Instrumentation for Fatigue Crack Detection. Sensors, 18(7), p.2161.
Round 2
Reviewer 1 Report
The authors have answered the required comments. After English writing modification, I would suggest to receive this manuscript published in this journal.
Author Response
AS FOR the 1st reviewer,
Thanks for the reviewer's comments. The entire manuscript has been carefully reviewed by native speakers.
Reviewer 2 Report
Thank you for addressing the comments. Please include details from the Response 1 in the manuscript.
